# Nonalcoholic Fatty Liver Disease in Patients with Type 2 Diabetes: Screening, Diagnosis, and Treatment

**DOI:** 10.3390/jcm12175597

**Published:** 2023-08-27

**Authors:** Stefano Ciardullo, Michela Vergani, Gianluca Perseghin

**Affiliations:** 1Department of Medicine and Rehabilitation, Policlinico di Monza, Via Modigliani 10, 20900 Monza, MB, Italy; m.vergani37@campus.unimib.it (M.V.); gianluca.perseghin@policlinicodimonza.it (G.P.); 2Department of Medicine and Surgery, University of Milano Bicocca, 20126 Milan, MI, Italy

**Keywords:** NAFLD, MASLD, GLP1-RA, SGLT2-I, fibrosis, screening, diabetes

## Abstract

Nonalcoholic fatty liver disease (NAFLD), recently renamed metabolic dysfunction-associated steatotic liver disease (MASLD) affects ~70% of patients with type 2 diabetes (T2D), with ~20% showing signs of advanced liver fibrosis. Patients with T2D are at an increased risk of developing cirrhosis, liver failure, and hepatocellular carcinoma and their liver-related mortality is doubled compared with non-diabetic individuals. Nonetheless, the condition is frequently overlooked and disease awareness is limited both among patients and among physicians. Given recent epidemiological evidence, clinical practice guidelines recommend screening for NAFLD/MASLD and advanced liver fibrosis in patients with T2D. While many drugs are currently being tested for the treatment of NAFLD/MASLD, none of them have yet received formal approval from regulatory agencies. However, several classes of antidiabetic drugs (namely pioglitazone, sodium-glucose transporter 2 inhibitors, glucagon-like peptide 1 receptor agonists, and multi-agonists) have shown favorable effects in terms of liver enzymes, liver fat content and, in some occasions, on histologic features such as inflammation and fibrosis. Therefore, diabetologists have the opportunity to actively treat NAFLD/MASLD, with a concrete possibility of changing the natural history of the disease. In the present narrative review, we summarize evidence and clinical recommendations for NAFLD/MAFLD screening in the setting of T2D, as well as on the effect of currently available glucose-lowering drugs on hepatic endpoints.

## 1. Epidemiology of NAFLD in Type 2 Diabetes

Nonalcoholic fatty liver disease (NAFLD), recently renamed metabolic dysfunction-associated fatty liver disease (MAFLD) [1] or metabolic dysfunction-associated steatotic liver disease (MASLD) [2], represents by far the most common chronic liver condition worldwide. Recent data showed that approximately one in three people from the general adult population [3,4] and one in four adolescents [5] is affected, making it one of the most common non-communicable diseases. Its development and progression are tightly linked with metabolic dysregulation and insulin resistance. It is therefore not surprising that its prevalence is even higher in patients with type 2 diabetes (T2D), where it reaches 60–75% [6,7,8]. Several studies have demonstrated the strong bidirectional relationship between NAFLD and T2D. On the one hand, NAFLD increases the risk of developing T2D among non-affected individuals [9] and the risk of micro- and macro-vascular complications in patients with previous T2D [10]; on the other, patients with diabetes tend to progress faster to its more advanced forms including nonalcoholic steatohepatitis (NASH, recently renamed metabolic dysfunction-associated steatohepatitis, MASH), advanced liver fibrosis, cirrhosis, and hepatocellular carcinoma [11].

In a recent study in patients with NAFLD/MASLD studied with paired liver biopsies, even after adjustment for potential confounders, the presence of T2D was associated with a 70% increase in the relative risk of fibrosis progression [12]. While previously considered a relatively uncommon finding, recent studies performed in unselected patients with T2D found that 15–38% of patients might have advanced liver fibrosis or cirrhosis (F3–F4) [6,8,13,14,15].

Importantly, the degree of liver fibrosis is the most important histologic predictor of future development of liver-related events, as shown by several cohort studies and meta-analyses [16,17]. The higher disease prevalence and the faster rate of progression account for the ~three times higher risk of dying from liver disease shown in patients with T2D compared with age- and sex-matched controls [18]. Nonetheless, awareness of this condition and its potential prognostic implications is limited both among affected individuals [19] and among healthcare professionals [20].

In the present narrative review, we will provide the reader with a practical summary of available evidence and guideline recommendations on screening, diagnosis, and treatment of NAFLD/MASLD in patients with T2D. We will also focus on the effect of currently available and future antidiabetic medication on liver-related endpoints.

## 2. Screening Strategies and Recommendations

Liver biopsy remains the established and most reliable method for assessing the severity of NAFLD/MASLD as it allows for a comprehensive evaluation of the three histologic aspects of the disease: steatosis, inflammation, and fibrosis. Nonetheless, this procedure is invasive and not well-received by patients [21] and it carries potential risks, including pain (30–50%) [22], severe bleeding (0.6%) [23] and, in rare cases, even death (up to 0.1%) [24]. Due to these drawbacks and its substantial costs, it is not suitable for widespread screening. Additionally, sampling errors are common since NAFLD can affect the liver in a non-uniform manner. For instance, Ratziu et al. reported that out of 51 patients who underwent two biopsies from the right lobe on the same day, 35% of those with bridging fibrosis (F3) in one sample were classified as having F0 or F1 in the other [25]. To overcome these limitations, several non-invasive methods were introduced to detect steatosis, inflammation, and fibrosis.

### 2.1. Identification of Liver Steatosis

International guidelines recommend the use of a conventional liver ultrasound as a first-line diagnostic technique to be applied in clinical practice for the diagnosis of steatosis (and therefore NAFLD/MASLD itself) and suggest that all patients with T2D, independently from liver enzymes levels, should be screened for steatosis with an ultrasound examination [26,27,28]. Even though it is an operator-dependent technique and its sensitivity might be limited in the setting of mild steatosis (involving 5–15% of hepatocytes) [29], accuracy is considered adequate for moderate-severe steatosis and it provides additional diagnostic information. While blood-based biomarkers (such as the Fatty Liver Index [30], the Hepatic Steatosis Index [31] and the NAFLD Liver Fat Score [32]) may be used in large epidemiologic studies to detect disease prevalence, their use as diagnostic methods in clinical practice is not recommended at the present time. In fact, these scores do not add much to the information provided by clinical, laboratory, and imaging examinations that are routinely performed in patients with suspected NAFLD/MASLD [27].

While magnetic resonance spectroscopy (MRS) or magnetic resonance imaging-Proton Density Fat Fraction (MRI-PDFF) have higher accuracy in identifying and quantifying intrahepatic fat compared with ultrasound [33], their use is limited to research studies and clinical trials due to their cost and limited availability.

Finally, Controlled Attenuation Parameter (CAP), a measure that can be obtained concomitantly with the liver stiffness measurement (LSM) using vibration-controlled transient elastography (VCTE), has gained attention as a rapid and inexpensive technique to detect and quantify liver fat [34]. The CAP value is expressed in dB/m and, even though different cut-offs have been proposed, a score ≥ 275 dB/m (derived from the study by Eddowes et al. [35]) is suggested by recent guidelines from the European Association for the Study of the Liver (EASL) [27]. However, given its limited availability and lack of head-to-head studies compared to ultrasound, the same guidelines do not recommend it as a first-line technique.

### 2.2. Identification of Liver Fibrosis

Given the prominent role of liver fibrosis in determining the prognosis of patients with NAFLD/MASLD and the difficulties encountered in the development of accurate biomarkers of inflammation, efforts have been made to identify accurate and readily available non-invasive methods to detect and stage liver fibrosis. Ideally, biomarkers should possess more than just diagnostic capabilities when compared to the current gold standard technique of liver biopsy. They should also have the ability to predict future liver-related events, facilitate disease monitoring over time, and indicate the effectiveness of a specific intervention [36]. Regrettably, such comprehensive biomarkers are not currently accessible, leading to significant ongoing efforts focused on their development and validation. Available noninvasive indicators of liver fibrosis fall into three categories: basic (non-patented) blood tests, specialized (patented) blood tests, and imaging methods. Table 1 lists most of these methods. Here, we will focus on the most validated and on those that are recommended by current clinical practice guidelines.

Nonproprietary algorithms utilizing common clinical and biochemical variables (indirect markers of fibrosis) are simple blood tests easily accessible in routine clinical practice, and they are correlated with histologic identification of advanced fibrosis. The most validated and applied are the FIB-4 [37] (based on AST, ALT, age, and platelet count) and the NAFLD Fibrosis Score [38] (NFS, based on AST, ALT, age, platelet count, BMI, and diabetes/impaired glucose tolerance). Both can be calculated by inserting these variables in calculators online. As shown in Table 2, their main advantages are the low cost, high availability, and the high negative predictive value (NPV, i.e., a FIB-4 value < 1.3 or a NFS < −0.675 are able to exclude the presence of advanced liver fibrosis with high probability). On the other hand, their positive predictive value (PPV) is limited and their performance is reduced in younger individuals. In a study involving >2700 patients with T2D, we showed that the agreement between these two biomarkers was low, as the proportion of patients with suspected advanced fibrosis ranged from 6.7% (FIB-4) to 33.3% (NFS) [39]. There is now general agreement that NFS tends to over-estimate the prevalence of advanced fibrosis in populations of patients with T2D or in the setting of bariatric surgery candidates [40]; this is due to the fact that both diabetes/impaired fasting glucose and BMI are part of the calculation of the score itself. We therefore suggest applying FIB-4 in the outpatient diabetes clinic.

Proprietary blood tests are designed with specific markers directly related to various elements of the extracellular matrix (ECM). In contrast to simple blood tests, these proprietary tests provide a more accurate representation of the dynamic processes involved in ECM deposition (fibrogenesis) and reorganization (fibrolysis). Many of these scores (listed in Table 1) rely on combinations of markers such as collagen III deposition (N-terminal propeptide PIIINP and Pro-C3), hyaluronic acid, α2-macroglobulin, and tissue inhibitor of matrix metalloproteinase 1 [41,42]. The most widely validated, and the only one recommended by clinical practice guidelines [28], is the Enhanced Liver Fibrosis test. As shown in Table 2, its performance is less affected by the age of the patient (its performance was high even in adolescents [43]) compared with simple tests, while its patented nature as well as higher costs reduce its availability in clinical practice. Importantly, Srivastava et al. showed that a two-step screening algorithm (in which ELF was performed in patients with a FIB-4 ≥ 1.3) was able to reduce unnecessary referrals of patients to hepatologists while increasing the ability to detect those with advanced liver fibrosis [44].

Finally, imaging technologies are generally based on elastography. Elastography methods capitalize on the principle that liver fibrosis results in increased liver stiffness, thereby altering the shear wave velocity and tissue displacement caused by ultrasound waves or a physical impulse [45]. These techniques can be categorized into two groups: ultrasound-based and magnetic resonance-based techniques. The advantages and disadvantages of these methods are summarized in Table 2.

Among the ultrasound-based techniques, vibration-controlled transient elastography (VCTE), commercially available as Fibroscan by Echosens, was the first to be introduced and is currently the most extensively validated [46]. Although it requires a dedicated device and it cannot provide the sonographer with the opportunity to visualize the liver and place the region of interest (ROI) in a location of his choice, it offers the advantage of simultaneously providing information on both liver stiffness measurement (LSM) and the degree of liver steatosis, which is estimated through the CAP.

Following that, elastography methods have been integrated into ultrasound devices using distinct technologies, such as point shear wave elastography (p-SWE) and two-dimensional shear wave elastography (2D-SWE) [47]. Despite some differences among these devices, their overall performance is similar [48] and generally, they outperform simple blood tests in identifying advanced fibrosis and cirrhosis [49]. However, these techniques have certain limitations, such as potential difficulties in obtaining measurements or producing invalid/unreliable results. Additionally, their accuracy is diminished in cases of acute hepatitis, cholestasis, recent food ingestion, congestive heart failure, and severe obesity [50,51].

Most of these limitations can be addressed through the application of magnetic resonance elastography (MRE). MRE employs specific hardware to generate a pulse sequence, and the acquired data are then processed by dedicated software to create a color elastogram of the entire liver [52]. Although MRE has less extensive experience compared to VCTE, studies focused on NAFLD/MASLD have demonstrated its superiority in identifying patients with advanced fibrosis and cirrhosis, consistently achieving AUROCs (Area Under the Receiver Operating Characteristic Curve) above 0.90 [53,54]. Due to these compelling findings, guidelines from the American Gastroenterological Association recommend MRE over VCTE for NAFLD patients suspected of having cirrhosis [55]. However, it is worth noting that MRE does have downsides, such as limited availability and higher costs, making it a technique most frequently used in clinical research in tertiary care centers rather than in clinical practice.

Several studies investigated whether combining either simple scores with patented ones or simple scores with imaging modalities could increase the accuracy in identifying the minority of patients with NAFLD/MASLD and advanced liver fibrosis, in order to facilitate referral to hepatologists for adequate management. For instance, a baseline analysis of the STELLAR trials, which evaluated the efficacy of selonsertib in patients with NASH, evaluated possible concomitant or sequential combinations of several noninvasive methods in a large population of patients evaluated by liver biopsy (*n* = 3202) [56]. The authors showed how FIB-4 followed by VCTE or ELF tests in those with indeterminate values (FIB-4 between 1.3 and 2.67) maintained an acceptable performance while reducing the rate of indeterminate results, therefore improving overall accuracy.

### 2.3. Proposed Screening Algorithms

The first guidelines to recommend generalized screening for NAFLD/MAFLD in patients with T2D were published in 2016 by EASL, the European Association for the Study of Diabetes (EASD) and the European Association for the Study of Obesity (EASO) [26]. This multi-society effort suggested generalized screening of NAFLD/MASLD in patients with T2D (but also in obese patients or those with metabolic syndrome) by performing conventional liver ultrasound. In case of steatosis, if liver enzymes were elevated, referral to the hepatologist was advised. If not, they recommended to calculate FIB-4 or NFS. If advanced fibrosis could be safely ruled out, re-evaluation could be performed in 2–3 years, while hepatologic referral was advised if this was not the case. Importantly, especially if liver enzymes are elevated, identification of other concomitant causes of advanced liver disease is recommended, including at least an evaluation of alcohol consumption and use of hepatotoxic medications as well as serology for viral hepatitis. An extended evaluation for the exclusion of less common conditions should be performed in a case-by-case manner.

While these guidelines represent a fundamental effort to achieve a systematic approach to the problem, several reports have underlined a potential for over-referral of patients in the setting of T2D or obesity clinics [40,57]. Moreover, lack of approved pharmacological therapies and few data on the cost-effectiveness of a similar screening strategy were considered as major areas of uncertainty. Indeed, these concerns were considered by the American Association for the Study of Liver Diseases’ (AASLDs’) 2018 guidelines, which did not recommend routine screening for NAFLD/MASLD even in patients at a high risk of disease progression, they advocated a case-finding strategy [58].

In the following years, data have accumulated on the performance of combining different methodologies and on the high prevalence of advanced fibrosis and cirrhosis in patients with T2D [6,8,44,56]. In 2021, EASL published an update to the clinical practice guidelines on non-invasive tests, in which a sequential algorithm similar to the one shown in Figure 1 was proposed.

The first step is FIB-4 calculation, and if a value ≥ 1.3 is obtained, VCTE should be performed. If the LSM value is ≥8 kPa, the patient should be referred to the hepatologist, while advanced fibrosis can safely be excluded if a lower value is obtained [27].

This strategy has generally been implemented in other recent guidelines from several international hepatologic and endocrinologic societies. In 2023, AASLD published new guidelines in which they recommended screening for advanced fibrosis in all patients with T2D, which was considered a condition facilitating progression towards cirrhosis [28]. Similar to the EASL guidelines, the first step is FIB-4, while, if a value ≥ 1.3 is obtained, the second step can be performed with VCTE, MRE, or ELF, depending upon availability. A very similar approach has also been recommended in recent guidelines from the American Association of Clinical Endocrinologists (AACE) [59] and by the 2023 Standards for Care in diabetes issued by the American Diabetes Association (ADA) [60]. The 2023 standards highlight the need for screening irrespectively of liver enzymes levels, as many patients with NAFLD/MASLD may have advanced fibrosis and ALTs/ASTs within the normal range [61].

As a summary, agreement on the need for systematic screening for NAFLD/MASLD, but most importantly for advanced liver fibrosis, has increased progressively in the last decade. Most international societies also recommend similar two-step algorithms in which a simple inexpensive blood test is performed first and more specialized tests are only performed in the case of uncertainty on the first. Our data on patients with T2D suggest that FIB-4 excludes advanced fibrosis in 55–60% of patients [39], leaving the remaining 40–45% to be studied with either VCTE or ELF.

## 3. Effect of Antidiabetic Drugs on NAFLD/MASLD

The important favorable effect of lifestyle interventions and bariatric surgery on histologic features of NAFLD/MASLD were demonstrated in multiple observational studies [62]. As a consequence, all international guidelines recognize the fundamental importance of nutrition, physical activity, and behavioral therapy in the long-term treatment of NAFLD/MASLD. These approaches, when successful in leading to significant weight loss, are associated with robust improvements in all histological aspects of the disease and require a multidisciplinary approach [62,63,64]. Nonetheless, NAFLD/MASLD is still considered an orphan disease in terms of pharmacologic therapies [65]. Indeed, while many drugs are being studied [66], none have yet received approval by pharmacological agencies with a specific indication to treat MASH.

Nonetheless, diabetologists are in a position of favor, as some drugs currently used to treat T2D itself have shown some efficacy on liver disease endpoints as well. Here, we report results with drug classes currently used for the treatment of T2D that were studied in RCTs on different NAFLD/MASLD endpoints. A summary of current evidence is also provided in Table 3, while Figure 2 shows the potential mechanisms underlying the hepato-protective effect of antidiabetic medications. Placebo-controlled RCTs examining the efficacy of sulphonylureas, acarbose, or insulin on NASH resolution, liver fat content and other liver function parameters are not available in the literature.

### 3.1. Pioglitazone

The rationale for employing pioglitazone is based on its activation of peroxisome proliferator-activated receptor gamma (PPARγ), which can exert a significant impact on the pathophysiology of NAFLD/MASLD. By ameliorating insulin resistance, modulating lipid and glucose metabolism in a favorable manner, and reducing hepatic and gastrointestinal inflammation, pioglitazone contributes to a reduction in portal hypertension, splanchnic inflammation, angiogenesis, and porto-systemic shunts [74].

Based on a post-hoc analysis involving 55 patients with biopsy-proven NASH, treatment with pioglitazone appears to result in changes in body fat distribution, specifically a decrease in the visceral-to-subcutaneous fat ratio, and biochemical alterations, including an increase in plasma adiponectin levels, when compared to the control group. These changes reflect the mechanism by which this drug contributes to the reduction of steatosis and necroinflammation in NASH patients [75].

In the literature there are several randomized controlled trials (RCTs) conducted in different parts of the world (USA, Europe, and Asia) which involve patients with or without T2D and biopsy-proven NAFLD/MASLD. A phase 2 meta-analysis published by Musso et al. encompasses eight RCTs, involving approximately 500 patients with biopsy-proven NASH, who were treated with thiazolidinediones [68]. These trials consist of five RCTs evaluating the use of pioglitazone and three RCTs evaluating the use of rosiglitazone, with treatment durations ranging from 6 to 24 months. Treatment with pioglitazone, compared to the control group (placebo or reference therapy), led to a higher proportion of patients reaching NASH resolution. Although individual studies did not show an improvement in fibrosis for any stage, when all studies were combined, thiazolidinedione therapy was associated with fibrosis improvement, fibrosis improvement at any stage, and resolution of NASH, even in patients without diabetes.

According to the European guidelines (EASL-EASD-EASO), pioglitazone may be used in the treatment of NAFLD/MASLD in individuals with type 2 diabetes (off-label in those without type 2 diabetes). On the other hand, the American guidelines (AASLD), in line with UK guidelines (NICE), suggest the use of pioglitazone only in diabetic individuals with NAFLD/MASLD [76]. The recent ADA guidelines, for the first time, specifically recommend certain classes of drugs for diabetic patients with the specific goal of improving NASH, and pioglitazone falls into one of those recommended classes [60].

### 3.2. GLP-1 RAs

The GLP-1 receptor agonists (GLP-1 RAs) play a direct and indirect role in the pathophysiology of NAFLD/MASLD. They increase insulin levels, thereby reducing hepatic gluconeogenesis. Additionally, GLP-1 RAs decrease lipolysis and the influx of free fatty acids in the liver. These agents may also contribute to reducing inflammation and apoptosis, promoting tissue remodeling, and increasing adiponectin levels. However, most importantly, they induce weight loss [77].

The first RCT conducted to evaluate the effect of a GLP1-RA on histologic liver endpoints was the Phase 2 LEAN trial, in which 26 patients were randomly assigned to receive liraglutide and 26 to placebo. This relatively small study showed positive results as nine (39%) of twenty-three patients who received liraglutide had NASH resolution, compared with two (9%) of twenty-two in the placebo group (relative risk 4.3 [95% CI 1.0–17.7]; *p* = 0.019). Moreover, two (9%) of twenty-three patients in the liraglutide group versus eight (36%) of twenty-two patients in the placebo group had progression of fibrosis (0.2 [0.1–1.0]; *p* = 0.04) [69].

The effectiveness of GLP-1 RAs has been recently summarized in a meta-analysis that included 11 placebo-controlled or active-controlled phase-2 RCTs involving a total of 936 middle-aged individuals [78]. These trials utilized different GLP-1 RAs specifically for the treatment of NAFLD/MASLD or NASH, diagnosed either by liver biopsy (in 2 RCTs) or imaging techniques (in 9 RCTs). Among all the trials, treatment with GLP-1 RAs was found to be safe, and significant reductions in hepatic fat were observed in RCTs with imaging-based endpoints. Moreover, RCTs with biopsy-based endpoints showed a significantly higher percentage of patients reaching NASH resolution compared to placebo. However, despite a trend, there were no statistically significant differences in fibrosis improvement. There could be several explanations for this, including a follow-up period that might have been too short to detect fibrosis improvement, a relatively long period of escalation to the maximum dose of GLP-1 RAs, an unreliable fibrosis assessment method due to the continuous nature of liver fibrosis, making it difficult to categorize, and reduced statistical power for the secondary endpoints. Due to these factors, further longer-term studies are required to definitively determine the effect of GLP-1 RAs on liver fibrosis. It is also important to highlight that the improvement in NASH was closely correlated with the reduction in body weight, which was, in turn, dose-dependent [69,70]. Therefore, it is valid to question whether the effect of GLP-1 RAs is solely mediated by weight loss. There is a line of research suggesting that GLP-1 RAs exert their effects by binding to a hepatocyte receptor. However, studies conducted on animals have demonstrated the scarcity of such receptors, making the mediated effect likely to be less significant. From the above-mentioned meta-analysis, a linear correlation is evident between the percentage reduction in hepatic fat content, assessed by MRI, and the reduction in BMI in patients treated with GLP-1 RAs, with an r^2^ value of 0.791. This supports the idea that the effect of GLP-1 RAs on NAFLD/MASLD endpoints is mostly driven by their capacity to promote weight loss [79]. Indeed, there are still limitations to consider in the use of GLP-1 RAs for NAFLD/MASLD treatment. Only two RCTs have utilized histological endpoints, and the data from Newsome et al. on semaglutide use may not be easily applicable to real-world clinical practice due to the different dosage used in their study compared to standard practice. Additionally, data on non-diabetic patients are still insufficient. Until recently, these drugs were only considered safe for the treatment of diabetic patients with NAFLD/MASLD, as indicated by various guidelines, without specific recommendations. However, with the publication of the recent ADA guidelines, GLP-1 RAs are now recommended for the first time as an adjunctive therapy to lifestyle interventions for adults with T2D, particularly those who are overweight or obese and have NAFLD/MASLD (Level of evidence B). Furthermore, GLP-1 RAs, along with pioglitazone, are denoted as the preferred agents for the treatment of hyperglycemia in adults with T2D and biopsy-proven NAFLD/MASLD or those at high risk for NAFLD/MASLD (Level of evidence A). These guidelines mark a significant advancement in the use of GLP-1 RAs in the management of NAFLD/MASLD in specific patient populations [60].

### 3.3. DPP4-i

The role of DPP4 inhibitors (DPP4-i) in NAFLD/MASLD is considered minimal. Although these drugs can reduce HbA1c levels, they have a neutral effect on weight. There are four placebo-controlled or active-controlled RCTs that used either sitagliptin (*n* = 3) or vildagliptin (*n* = 1) to specifically treat NAFLD/MASLD [67]. These RCTs, in which NAFLD/MASLD was detected by imaging techniques, took place in different parts of the world including Europe, Asia, and the USA. When compared to placebo or reference therapy, vildagliptin had a marginally significant beneficial effect on liver fat and showed a mild reduction in serum ALT levels, whereas sitagliptin did not. Given the absence of liver histological data, we are unable to comment on the effect of DPP-4 inhibitors on the histological improvement of NAFLD/MASLD. International guidelines agree in considering DPP4-I safe for the treatment of patients with T2D and liver disease.

### 3.4. SGLT2-i

SGLT2 inhibitors (SGLT2-I) induce a series of modifications that have the potential to improve the liver condition in individuals with NAFLD/MASLD: they improve glycemic profile, reduce visceral adipose tissue, increase plasma adiponectin levels, and decrease uric acid levels; they also diminish oxidative stress and systemic inflammation and increase glucacon levels [80].

Twelve RCTs involving the use of these drugs for the specific treatment of NAFLD/MASLD were conducted in different parts of the world (Asia, Europe, and the United States) [71]. The SGLT2-I considered in the various RCTs include dapagliflozin (*n* = 6 RCTs), empagliflozin (*n* = 3 RCTs), ipragliflozin (*n* = 2 RCTs), and canagliflozin (*n* = 1 RCT), administered for a median period of 24 weeks. The subjects included in these studies were predominantly diabetic (≃90%), and NAFLD/MASLD was diagnosed through imaging. From the meta-analysis of these studies, it is evident that SGLT2-I, compared to the control group (placebo or reference therapy), led to a significant reduction in the percentage of hepatic fat as evaluated by MRI. Currently, there are no published RCTs that utilize histological hepatic endpoints. According to most international guidelines, SGLT2-I are considered safe for patients with liver diseases, but their specific use for the treatment of NAFLD/MASLD is not yet recommended.

### 3.5. Metformin

Metformin is the first-line drug in the treatment of diabetes and can reduce gluconeogenesis and liponeogenesis, lower systemic inflammation, increase GLP1 levels, and modify intestinal microbiota, all of which have potential beneficial effects on the liver [81].

In the literature, there are trials conducted in various parts of the world that involve the use of metformin in diabetic, non-diabetic, or prediabetic patients with NAFLD/MASLD diagnosed through biopsy or imaging. It has been observed that metformin can reduce liver steatosis, leading to a significant decrease in transaminase levels (especially ALT). However, it does not have a significant effect on fibrosis and resolution of NASH [67]. The AISF-SID-SIO 2021 guidelines consider metformin safe for diabetic patients with liver disease, but there is no specific indication for the treatment of NAFLD/MASLD [76].

### 3.6. Double and Triple Incretin-Receptor Agonist

Tirzepatide, a dual glucose-dependent insulinotropic polypeptide (GIP) and GLP-1 receptor agonist, has demonstrated superiority compared with GLP1-RA on both glycemic control and weight loss, and achieved favorable effects on hepatic endpoints. In a substudy of the SURPASS-3 trial, which enrolled exclusively patients with T2D, tirzepatide showed a significantly greater reduction in liver fat content (LFC), volume of visceral adipose tissue (VAT), and abdominal subcutaneous adipose tissue (ASAT) compared to insulin degludec [72]. These results are promising, yet to date no data are available on the effect of tirzepatide on histologic endpoints.

Retatrutide (RETA), a once-weekly injectable triple hormone agonist of the GIP, GLP-1, and glucagon receptors, has also shown promise in the treatment of obesity. In a phase 2 obesity trial, RETA treatment resulted in substantial reductions in body weight (up to 24%) [82]. In a substudy of the same trial involving participants with NAFLD/MASLD, all doses of RETA showed significantly greater reductions in liver fat content compared to placebo. RETA doses of 8 and 12 mg led to hepatic steatosis resolution (LFC < 5%) in more than 85% of participants at week 48 [73]. RETA also demonstrated improvements in some NASH biomarkers, such as K-18 and Pro-C3. The efficacy of these novel drugs in individuals with NAFLD/MASLD is promising and supports further evaluation to establish their treatment indications in this patient population.

Recommendations of international societies on screening and pharmacological treatment of NAFLD/MASLD in patients with T2DM are summarized in Table 4.

## 4. Conclusions

NAFLD/MAFLD and T2D are tightly linked in a strong, bidirectional relationship. Given the extremely high disease prevalence, referral of all affected patients to hepatologists is not feasible or cost-effective. Diabetologists are therefore called to actively screen and risk-stratify patients to identify those that are at higher risk of clinically relevant liver-related outcomes. Current guidelines recommend a two-step strategy in which a simple blood-based score such as FIB-4 is followed (if advanced fibrosis cannot directly be excluded) by an imaging technique (most commonly VCTE). This screening procedure does not only aim to identify patients to refer to the hepatology clinic, but also to inform treatment. Indeed, while no specific agent has been approved with the indication to improve NAFLD/MASLD outcomes, several glucose-lowering agents showed some efficacy on hepatic endpoints in dedicated RCTs. In particular recent guidelines recommend pioglitazone or GLP1-RA in patients with T2D and biopsy-proven NASH or those at a high risk of advanced liver fibrosis. We believe that diabetologists are currently in a privileged position to actively treat patients with T2D not only to reduce their risk of developing micro- and macro-vascular complications, but also to reduce the disease burden associated with cirrhosis and hepatocellular carcinoma.

## Figures and Tables

**Figure 1 jcm-12-05597-f001:**
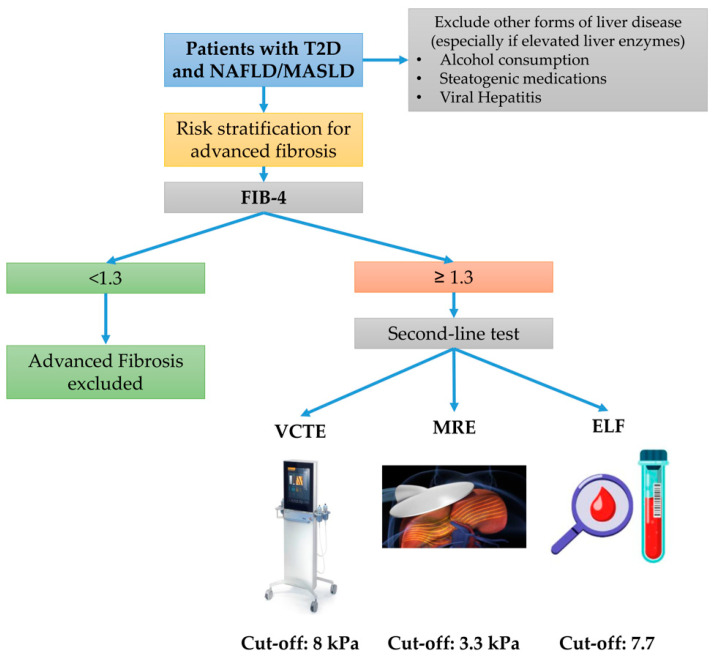
Recommended screening algorithm for patients with type 2 diabetes and NAFLD/MASLD. Abbreviations: NAFLD, Nonalcoholic Fatty Liver Disease; MASLD, metabolic dysfunction-associated steatotic liver disease; FIB-4, Fibrosis 4 index; VCTE, Vibration-Controlled Transient Elastography; MRE, Magnetic Resonance Elastography; and ELF, Enhanced Liver Fibrosis test.

**Figure 2 jcm-12-05597-f002:**
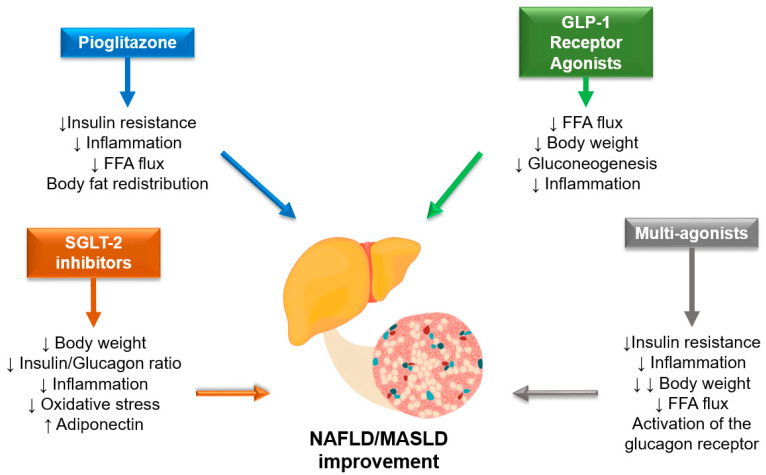
Potential mechanisms underlying the hepato-protective effect of selected glucose-lowering medications. Abbreviations: NAFLD, Nonalcoholic Fatty Liver Disease; MASLD, metabolic dysfunction-associated steatotic liver disease; SGLT2-I, Sodium Glucose Transporter 2 inhibitors; GLP1, glucagon-like peptide 1; and FFA, free fatty acids. ↓: decrease; ↑: increase.

**Table 1 jcm-12-05597-t001:** Available non-invasive techniques to estimate the degree of liver fibrosis in patients with NAFLD/MASLD.

Non-Patented Blood Tests	Patented Blood Tests	Imaging Modalities
AST to platelet ratio index (APRI)	Enhanced Liver Fibrosis (ELF)	Vibration Controlled Transient elastography (VCTE)
Fibrosis-4 Index (Fib-4)	Fibrometer	Point share wave elastography/ARFI
NAFLD fibrosis score (NFS)	Fibrotest	2D share wave elastography
Hepamet fibrosis score (HFS)	Pro-C3	Magnetic resonance elastography (MRE)

Abbreviations: ALT, alanine aminotransferase; AST, aspartate aminotransferase; ARFI, Acoustic Radiation Force Impulse; MASLD, Metabolic dysfunction-associated fatty liver disease; and NAFLD, Nonalcoholic fatty liver disease.

**Table 2 jcm-12-05597-t002:** Diagnostic features of noninvasive methods recommended for the identification of advanced liver fibrosis in patients with NAFLD/MASLD.

Test	Strengths	Weaknesses	Additional Considerations
Serum biomarkers			
Fibrosis-4 Index (Fib-4)	Cheap and easy to perform (based on age, AST, ALT, and platelet) High NPV	Low PPV Influenced by other causes of low platelet levels Large indeterminate area	Age-adjusted cut-offs may be considered in older patients (≥65 years) Low accuracy in younger patients (<35 years)
NAFLD fibrosis score (NFS)	Cheap and easy to perform (based on age, AST, ALT, platelet, BMI, diabetes, and albumin) High NPV	Low PPV Influenced by other causes of low platelet levels Large indeterminate area	Overestimates fibrosis if applied to a population of patients with T2D or in the setting of severe obesity
Enhanced Liver Fibrosis (ELF)	Easy to perform High NPV Performance not affected by age	Patented and therefore more expansive and less available Indeterminate area	Based on the measurement of type III procollagen peptide (PIIINP), hyaluronic acid (HA), and tissue inhibitor of metalloproteinase-1 (TIMP1)
**Imaging techniques**			
Vibration Controlled Transient Elastography (VCTE)	Most widely validated technique Fast and easy to perform Prognostic value	Dedicated device needed Not yet widely available Does not allow direct visualization of the liver parenchima for ROI placement	Performance affected by acute inflammation, severe obesity, ascitis, cholestasis, food intake nd heart failure Well-validated quality criteria
Point Share Wave Elastography (SWE) and 2-Dimensional SWE	Can be performed in combination with regular ultrasound Operator can choose the ROI Can be performed in case of ascites	Performance affected by acute inflammation, severe obesity, ascitis, cholestasis, food intake nd heart failure Less validated compared with VCTE due to more recent introduction	Softwares can be installed on many new-generation ultrsound machines Few data on prognostic ability
Magnetic Resonance Elasography (MRE)	Examines the whole liver Better performance compared with ultrasound-based techniques Not affected by ascites or obesity	Costly and time consuming Requires an MRI facility Very limited availability	Considered by some the “gold standard” noninvasive technique to identify liver fibrosis AUCs ≥ 0.90 for advanced fibrosis and cirrhosis compared with liver biopsy

Abbreviations: AST, aspartate aminotransferase; ALT, alanine aminotransferase; NAFLD, Nonalcoholic fatty liver disease; AUC, Area Under the Curve; MRI, Magnetic Resonance Imaging; and ROI; Region of Interest.

**Table 3 jcm-12-05597-t003:** Summary of results obtained with glucose-lowering medications on NAFLD/MASLD.

	RCT’s Characteristics	AST, ALT	Liver Fat Content	Liver Fibrosis	NASH Resolution
Metformin [67]	Biopsy- and imaging-proven NAFLD; diabetic, prediabetic and non diabetic adults; children and adolescents with metabolic disfunctions	Improved	Improved	No effect	No effect
DPP4-i [67]	Imaging-proven NAFLD; diabetic and prediabetic adults	Marginally (vidagliptin)	Marginally (vidagliptin)	Unknown	Unknown
Pioglitazione [68]	Biopsy-proven NAFLD; diabetic, prediabetic and non diabetic adults	Improved	Improved	Improved (meta-analysis)	Improved
GLP-1 RAs [69,70]	Biopsy- and Imaging- proven-NAFLD; mostly diabetic but also non diabetic asdults	Improved	Improved	No effect	Improved
SGLT2-i [71]	Imaging-proven NAFLD; mostly diabetic but also non diabetic adults	Improved	Improved	Unknown	Unknown
Tirzepatide [72]	Imaging-proven NAFLD; diabetic adults	Improved	Improved	Unknown	Unknown
Retatrutide [73]	Imaging-proven NAFLD	No effect	Improved	Unknown	Unknown

Abbreviations: AST, aspartate aminotransferase; ALT, alanine aminotransferase; GLP1-RA, glucagon-like peptide 1 receptor agonists; and SGLT2-I, sodium-glucose transporter 2 inhibitors.

**Table 4 jcm-12-05597-t004:** Recommendations for screening and pharmacological treatment of NAFLD/MASLD in patients with type 2 diabetes according to international societies.

Society	Screening	Pharmacological Treatment
AASLD [28]	All patients with hepatic steatosis or clinically suspected NAFLD based on the presence of obesity and metabolic risk factors should undergo primary risk assessment with FIB-4. High-risk individuals, such as those with T2D, medically complicated obesity, family history of cirrhosis, or more than mild alcohol consumption, should be screened for advanced fibrosis. If FIB-4 is ≥1.3, VCTE, MRE, or ELF may be used to exclude advanced fibrosis.	Semaglutide can be considered for its approved indications (T2D/obesity) in patients with NASH, as it confers a cardiovascular benefit and improves NASH. Pioglitazone improves NASH and can be considered for patients with NASH in the context of patients with T2DM.
EASL [26,27]	In patients with T2D, the presence of NAFLD should be looked for irrespective of liver enzyme levels, since T2D patients are at high risk of disease progression. In patients with NAFLD, the following NITs are recommended to rule-out advanced fibrosis in clinical practice (LoE 1, strong recommendation): - LSM by TE < 8 kPa; - ELF < 9.8 or FibroMeterTM <0.45 or FibroTest^®^ < 0.48; - FIB-4 < 1.3 or NFS < −1.455	While no firm recommendations can be made, pioglitazone (most efficacy data, but off-label outside T2D) or vitamin E (better safety and tolerability in the short-term), or their combination could be used for NASH.
AACE [59]	In persons with T2D, clinicians should consider screening for clinically significant fibrosis (stages F2–F4) using the FIB-4, even if they have normal liver enzyme levels. Clinicians should consider persons belonging to the “high-risk” groups who have an indeterminate or high FIB-4 score for further workup with an LSM (transient elastography) or ELF test, as available.	Pioglitazone and GLP-1 Ras are recommended for persons with T2D and biopsy-proven NASH. Clinicians must consider treating diabetes with pioglitazone and/or GLP-1 Ras when there is an elevated probability of having NASH based on elevated plasma aminotransferase levels and noninvasive tests.
APASL [83]	Screening for MAFLD by ultrasonography should be considered in at-risk populations such as patients with overweight/obesity, T2D, and metabolic syndrome. The exclusion of high risk of significant or advanced fibrosis is acceptable using non-invasive tools, liver stiffness measurement by VCTE, or shear wave elastography and blood biomarkers and scores of fibrosis or their sequential combination.	No formal recommendations on pharmacological treatment of NAFLD/MAFLD/MASLD.
ADA [60]	Adults with T2D or prediabetes, particularly those with obesity or cardiometabolic risk factors/established cardiovascular disease, should be screened/risk stratified for NAFLD with significant liver fibrosis using a calculated FIB-4 index, even if they have normal liver enzymes. Adults with T2D or prediabetes with an indeterminate or high FIB-4 index should have additional risk stratification by LSM with TE or the blood biomarker ELF.	For adults with T2D, particularly with overweight or obesity with NAFLD, consider using a GLP1-RA with demonstrated benefits in NASH as an adjunctive therapy to lifestyle interventions for weight loss. Pioglitazone or GLP1-RA are the preferred agents for the treatment of hyperglycemia in adults with T2DM with biopsy-proven NASH, or those at high risk for NAFLD with significant liver fibrosis using noninvasive tests.

## Data Availability

No new data were created as the present was a narrative review.

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
