# Peer review of "Nonalcoholic Fatty Liver Disease in Patients with Type 2 Diabetes: Screening, Diagnosis, and Treatment"

_jcm, 2023, doi:10.3390/jcm12175597_

Round 1

Reviewer 1 Report

Ciardullo et al. summarized the literature for a topic review on non-alcoholic fatty liver disease (NAFLD) in patients with type 2 diabetes. This review emphasized screening, diagnosis, and antidiabetic medications that influence NAFLD. While the topic is intriguing, some aspects require clarification.

Major comments:

1.     Could the authors summarize the screening recommendation and tools according to the AASLD, EASL, APASL, or AACE in the table?

2.     In Table 3, please add the references and detail of the studies.

3.     Please add the table or figure that guides the antidiabetic medications for patients with NAFLD.

4.     Figure Explaining the Mechanisms: I'd recommend a flowchart or infographic that illustrates how various antidiabetic medications influence the pathogenesis of NAFLD. This can include pathways like insulin resistance modulation, effects on liver fat accumulation, anti-inflammatory actions, etc.

Author Response

Ciardullo et al. summarized the literature for a topic review on non-alcoholic fatty liver disease (NAFLD) in patients with type 2 diabetes. This review emphasized screening, diagnosis, and antidiabetic medications that influence NAFLD. While the topic is intriguing, some aspects require clarification.

Major comments:

  1. Could the authors summarize the screening recommendation and tools according to the AASLD, EASL, APASL, or AACE in the table?

We thank the Reviewer for His/Her comment. We included a new Table (Table 4) showing recommendations from the mentioned societies on both screening and treatment of NAFLD/MASLD in patients with type 2 diabetes.

  1. In Table 3, please add the references and detail of the studies.

We thank the Reviewer for His/Her comment. We included references supporting statements shown in Table 3.

  1. Please add the table or figure that guides the antidiabetic medications for patients with NAFLD.

We thank the Reviewer for His/Her comment. Rather than including a new Figure showing therapeutic algorithms for T2D, which are quite complex, we included recommendations given by international societies both on screening and treatment of NAFLD/MASLD (new Table 4).

  1. Figure Explaining the Mechanisms: I'd recommend a flowchart or infographic that illustrates how various antidiabetic medications influence the pathogenesis of NAFLD. This can include pathways like insulin resistance modulation, effects on liver fat accumulation, anti-inflammatory actions, etc.

We thank the Reviewer for His/Her comment. As suggested, we included such a Figure (Figure 2). We thank the Reviewer for His/Her suggestion as we believe that this addition significantly improved the overall quality of the manuscript.

Reviewer 2 Report

NAFLD/MAFLD is one of the most prevalent chronic liver diseases, and its associated problems are significant issues. This paper summarizes the evidence for NAFLD/MAFLD screening, diagnosis, and treatment in the context of type 2 diabetes. The primary contribution of this paper is an updated review that includes the impact of antidiabetic medications on liver-related outcomes. 

 1.         Table 3. Summary of results obtained with glucose-lowering medications on NAFLD/MASLD. In the second column (RCT characteristics), various RCTs have different sources. Please provide references accordingly.

2.         Some evidence indicates that lifestyle change, such as nutrition, physical activity, and behavioral therapy, is important for patients with diabetes and beneficial for NAFLD. I recommend adding some relevant literature to the article to give a more comprehensive description of the treatment.

Author Response

NAFLD/MAFLD is one of the most prevalent chronic liver diseases, and its associated problems are significant issues. This paper summarizes the evidence for NAFLD/MAFLD screening, diagnosis, and treatment in the context of type 2 diabetes. The primary contribution of this paper is an updated review that includes the impact of antidiabetic medications on liver-related outcomes.

  1. Table 3. Summary of results obtained with glucose-lowering medications on NAFLD/MASLD. In the second column (RCT characteristics), various RCTs have different sources. Please provide references accordingly.

We thank the Reviewer for His/Her comment. We included references supporting statements shown in Table 3.

  1. Some evidence indicates that lifestyle change, such as nutrition, physical activity, and behavioral therapy, is important for patients with diabetes and beneficial for NAFLD. I recommend adding some relevant literature to the article to give a more comprehensive description of the treatment.

We thank the Reviewer for His/Her comment. We agree on the importance of lifestyle changes in the management of NAFLD/MASLD and advanced fibrosis. Therefore, we expanded the section related to these aspects and included some new references.

Round 2

Reviewer 1 Report

Ciardullo et al. addressed all issues clearly. This review is interesting for readers on the topic of NAFLD in screening, diagnosis, and the influence of antidiabetic medications on NAFLD.